# Rochelle Salt-Based Ferroelectric and Piezoelectric Composite Produced with Simple Additive Manufacturing Techniques

**DOI:** 10.3390/ma14206132

**Published:** 2021-10-15

**Authors:** Etienne Lemaire, Damien Thuau, Jean-Baptiste De Vaulx, Nicolas Vaissiere, Atli Atilla

**Affiliations:** 1GREMAN, UMR-CNRS7347, Polytech Tours, Université de Tours, F-37200 Tours, France; 2LabECAM, ECAM Lyon, Université de Lyon, F-69005 Lyon, France; j.de-vaulx@ecam.fr (J.-B.D.V.); nicolas.vaissiere@ecam.fr (N.V.); atilla.atli@ecam.fr (A.A.); 3IMS, UMR5218, ENSCBP, CNRS, Bordeaux INP, Université de Bordeaux, F-33607 Pessac, France; damien.thuau@ims-bordeaux.fr

**Keywords:** 3D printing, epitaxy, ferroelectric, piezoelectric salt, eco-friendly, composite

## Abstract

More than one century ago, piezoelectricity and ferroelectricity were discovered using Rochelle salt crystals. Today, modern societies are invited to switch to a resilient and circular economic model. In this context, this work proposes a method to manufacture piezoelectric devices made from agro-resources such as tartaric acid and polylactide, thereby significantly reducing the energy budget without requiring any sophisticated equipment. These piezoelectric devices are manufactured by liquid-phase epitaxy-grown Rochelle salt (RS) crystals in a 3D-printed poly(Lactic acid) (PLA) matrix, which is an artificial squared mesh which mimics anatomy of natural wood. This composite material can easily be produced in any fablab with renewable materials and at low processing temperatures, which reduces the total energy consumed. Manufactured biodegradable samples are fully recyclable and have good piezoelectric properties without any poling step. The measured piezoelectric coefficients of manufactured samples are higher than many piezoelectric polymers such as PVDF-TrFE.

## 1. Introduction

The discovery of the piezoelectric effect is now more than one century ago. Piezoelectricity consists of converting mechanical energy into electrical energy; there is a direct piezoelectric effect when a stress is applied to a sample. Alternatively, an electrical-to-mechanical conversion happens when the voltage is applied to the sample, i.e., the converse piezoelectric effect. One century ago, Nicolson designed Rochelle salt piezoelectric transducers [1]. Since then, a large number of natural [2] and synthetic materials [3,4] have been reported with heterogeneous piezoelectric properties for numerous applications, including sensors, motors, actuators, and energy harvesters [5,6]. Most piezoelectric materials are inorganic ferroelectric perovskites, such as Lead Zirconate Titanate (PZT) and Barium Titanate (BTO). Environmental incompatibilities (toxicity, high sintering temperatures, etc.), as well as complex manufacturing procedures, have not prevented the widespread use of these ceramic materials, which are extensively employed in industry. They are still used due to their very high ferroelectric and piezoelectric coefficients. As a matter of fact, researchers have intended to continuously optimize these features over the years, e.g., by modifying their compositions [7,8]. Nowadays, piezoelectric organic ferroelectrics are attracting increased attention, but some organic ferroelectrics can be harmful to human health [9]. Among them, single crystals of croconic acid [10] have shown remanent polarizations of hundreds of µC·m^−2^. Piezoelectric constants *d*_33_ of nearly 110 pC·N^−1^ have been measured in polyamide 11/NaNbO_3_ nanowire composites [11]. A coefficient of 40 pm·V^−1^ has been recorded for thin layers of imidazolium perchlorate [12,13], and values greater than −60 pm·V^−1^ have been obtained from devices of polyvinylidene-fluoride-trifluoroethylene, P(VDF–TrFE) [14,15], and other organic crystals or materials [16,17], some of which could be 3D-printed [18]. Even if new piezoelectric materials are continuously discovered, traditional materials such as PZT, BTO and quartz are still studied [19] and used in numerous applications, while the oldest ones have surprisingly been somewhat abandoned. One such material, potassium sodium tartrate tetrahydrate (KNaC_4_H_4_O_6_.4H_2_O), also known as Seignette or Rochelle salt (RS), is ferroelectric between −18 °C and 24.9 °C (its two Curie temperatures) [20]. RS exhibits high piezoelectric and dielectric constants and has a monoclinic structure in the ferroelectric phase and an orthorhombic structure in the paraelectric phase [21], and is also soluble in water and non-toxic (RS: E337 food additive is approved by the Food and Drug Administration). Its low cost and ease of synthesis are some of its additional assets. It exhibits a piezoelectric coefficient that can be greater than 100 pm·V^−1^ [22]. Nevertheless, direct piezoelectric coefficients and coupling factors have not been recently reported due to the insights provided by new performant apparatus. This lack of interest might be due to its depreciated performance regarding its temperature and stability issues. Thus, this material has been excluded from technological applications over the past century. Nevertheless, from an environmental perspective, RS has some advantages in terms of its ease of production, processing simplicity, biocompatibility and biodegradability, resource scarcity, and intrinsic piezoelectric coefficient compared to most lead-based and lead-free piezoelectric elements. Additionally, some cooperatives still produce RS as a by-product of the wine industry and thus it can be considered a renewable agro-resource. Recently, we have shown the possibility of growing RS crystals in cellulose-based sheets [23] or in the naturally orientated capillaries of wood [24] to create eco-friendly piezoelectric transducers.

Here, an innovative perspective based on this green material is presented which consists of growing crystalline salt in a 3D-printed biodegradable polymer matrix, making its manufacture feasible in any fablab. We have developed 3D-printed, fully biodegradable piezoelectric salt composites. After the additive manufacturing of the PLA matrix, the liquid-phase epitaxial growth of RS crystals is performed with a controlled crystalline orientation. The resulting 3D-printed PLA matrix imitates the natural tubular structure of wood which has been recently studied [24]. The biodegradable PLA matrix has mechanical properties which reinforce the salt-based composite by encapsulating the brittle RS crystals. The millimetric mesh size of the 3D-printed PLA matrix favors the growth of single crystals with the same crystal orientation, thereby improving the piezoelectric performance.

## 2. Results and Discussion

Herein, a simple approach is implemented where disk-shaped piezoelectric samples composed of a 3D-printed PLA matrix and RS were manufactured by following the process described in the Experimental Section (see Section 4 and Figure 1). The manufactured samples were characterized by using direct and converse piezoelectricity measurements associated with impedance spectroscopy in order to illustrate the resonant device behavior and ferroelectricity.

### 2.1. X-ray Diffraction (XRD) Data

Comparative X-ray Diffraction experiments were performed on pure PLA matrix, RS-incorporated PLA matrix, roughly grinded single crystals of RS and commercial RS powder (Figure 2).

Wide-angle X-ray Diffraction experiments were performed with a co-X-ray source. Concerning the neat PLA, a main broad peak at around 18–19° and a shoulder around 37° were recorded, indicating an amorphous crystal structure (Figure 2a). A similar diffractogram has also been observed [25].

The commercial RS powder exhibited a complex XRD diffractogram with several diffraction patterns (Figure 2d). A main peak at around 18.78° was observed, as also detected by Shyju et al. [26], who attributed it to the (210) diffraction planes. A previously grown RS crystal was roughly grinded and the obtained small crystals were analyzed in XRD, which also showed a main pattern at around 18.7°, with some additional peaks at higher angles (Figure 2c). When RS crystals were grown in the PLA matrix by liquid-phase crystal growth technique, in addition to the broad XRD peaks originated from the PLA matrix, the peaks from RS crystals were also detected (Figure 2b). This observation indicates that RS crystals with specific orientations were grown in the PLA matrix cells.

### 2.2. Ferroelectricity

In order to measure the effective ferroelectric behavior of devices, the capacitance under polarization was measured (Figure 3). A square-shaped (10 mm side and 1 mm thick) pure RS crystal and a PLA/RS composite disk (28 mm diameter and 1 mm thin) were poled from −40 V to +40 V (and from +40 V to −40 V in order to highlight hysteresis loop) and capacitance at 10 kHz was measured every 200 mV (ten times). The impedance of the RS is strongly dependent on the temperature, with up to a 10% impedance variation per degree Celsius [27]. A precise temperature control (+/−0.1 °C) was necessary in order to observe ferroelectric behavior with minimized temperature effects. Thus, impedance variation with respect to polarization was varied in the same range (10% of impedance variation for the full scale of bias voltage available: 40 V). A pure RS sample has a capacitance variation from 65 to 85 pF during the polarization cycle (and an impedance from 0.9 to 1.2 MΩ with bias voltage varying from −40 V to +40 V). The PLA/RS disk displays a similar behavior with a capacitance variation of 116 to 128 pF during the polarization cycle. These measurements were performed near to 10 °C and similar curves have been obtained when scaled up to 15 °C. The typical hysteresis loop can be observed during the capacitance measurement of both devices (Figure 3). Therefore, the ferroelectric characterization of Rochelle salt established in 1947 [28] and afterwards [29,30] was only partially reproduced. The few experimental literature sources on Rochelle salt ferroelectricity suggest that we could reinvestigate Rochelle salt ferroelectric specificities under several environmental conditions (temperature and humidity) with modern equipment and a more sophisticated and dedicated experimental setup.

### 2.3. Direct Piezoelectricity

In order to measure the effective *d*_33_ coefficient, a controlled force step was applied to the sample and the sample output current was recorded (Figure 4). Then, the charges were integrated in order to obtain the *d* (pC·N^−1^) coefficient by using the known applied effective force. For the same applied force, most of the composite samples were responding with a higher electrical current than commercial PVDF-TrFE samples (MEAS piezoelectric film provided by TE connectivity, dimensions: 41 mm × 16 mm × 40 μm) loaded on 1 MΩ.

The *d*_33_ coefficients (Table 1) were calculated from electrical charges harvested under the same pressure step of 0 to 1 bar for manufactured PLA/RS composite samples of two different thicknesses and for a commercial PVDF-TrFE sample. The results show that RS/PLA samples had higher *d*_33_ coefficients under this amount of pressure. The commercial PVDF-TrFE had a reproducible *d*_33_ of 22 pC·N^−1^. Notice that its datasheet stated a *d*_33_ of 23 pC·N^−1^. The manufactured 2.5 mm and 1.5 mm thicknesses of RS/PLA samples had average *d*_33_ values of 39 pC·N^−1^ and 119 pC·N^−1^, respectively, under the same ambient conditions. Therefore, our thicker, manufactured composite disks have surprisingly high piezoelectric characteristics compared to commercial ultra-thin PVDF-TrFE. In our recent work, where RS crystals were grown in a natural wood matrix structure, an average value of 11 pC·N^−1^ for *d*_33_ was found [24]. It is worth noting that some thin PLA/RS samples (thicknesses of 1.5 mm) have non-negligible residual stress due to the 3D-printing process, resulting in a curved matrix shape rather than a flat sample. A stronger electrical response was thus recorded in this case due to the fact that the mechanical deformation includes not only pure compression but also the deflection of the disk. Therefore, the piezoelectricity in other directions is probably induced and other coefficients such as *d*_13_ and *d*_14_ are probably solicited. Note that those coefficients could not be measured on our samples. The average coefficients are included in a large uncertainty interval for RS/PLA samples due to reproducibility of the manufacturing process. In fact, crystallizations having defects of 3D-printed PLA matrices and residual stress issues led to samples with varying performances. Nevertheless, these results were confirmed by converse piezoelectricity measurements. Injected PLA matrices with a clean surface state could enhance the reproducibility.

### 2.4. Converse Piezoelectricity

AC signals were applied to the manufactured RS/PLA composite samples, and their displacements and electrical characteristics were recorded with a laser vibrometer and impedance analyzer. Both characterizations highlighted the same converse piezoelectric effect. For instance, the same sample measured with both techniques (Figure 5) showed the same principal resonant mode on conductance and out-of-plane displacement. The converse effective *d*_33_ (pm·V^−1^) coefficient was calculated by measuring the static displacement of the sample using the second-order approximation of the spring-mass-damper system given by:(1)$$H_{0}=2\xi H\left(f_{0}\right)$$
where $H_{0}$ is the static displacement, ξ is the damping ratio and $H\left(f_{0}\right)$ is the maximum displacement at the eigenfrequency $f_{0}$. The static displacements were calculated for several applied voltages U, oscillating at $f_{0}$, and reported (Figure 6). The displacement was found to increase linearly with the applied voltage. The slope of this linear function leads to the experimental effective converse *d*_33_ coefficient. The comparison of converse piezoelectric coefficients calculated from static displacement and applied voltage, for two thicknesses of RS/PLA manufactured samples (Table 1), confirmed the general trend measured for direct piezoelectric effect characterization. Fully biodegradable RS/PLA samples of 2.5 mm thick have an average *d*_33_ of 31 pm·V^−1^ and 1.5 mm thick RS/PLA samples have an average value of 80 pm·V^−1^. Under a laser vibrometer, displacement is measured in the same direction of the applied electric field (more consistent *d*_33_ measurement), and the sample residual stress is mostly eliminated from the sample out-of-plane response (measured only in this direction). Therefore, the coefficient calculated from converse piezoelectricity for thin samples is lower compared to the average one measured under mechanical solicitation. Additionally, the coefficient from converse piezoelectricity for thick PLA/RS samples is closer to the one obtained with direct piezoelectricity. This should be due to the absence of residual stress in thicker samples.

Thicker samples have lower effective *d*_33_ coefficients than thinner ones; therefore, it is important to clarify the thickness-dependence of effective *d*_33_ coefficients. Theoretically, liquid-phase epitaxy permits crystal growth in only one crystallographic direction which is the seed-crystal direction (homogenous growth). Nevertheless, in our case, the crystallization can also be initiated on the walls of the PLA matrix due to the defects on its rough surface (heterogeneous growth). Thus, as the crystallization progresses without facing the imperfect PLA matrix surface, the initial seed crystallization direction is further conserved. The increased contact surface in thicker PLA matrices increases other potential nucleation points for RS crystal growth on PLA walls. This is then a competition between homogenous and heterogeneous crystal growth. Therefore, in thicker samples, the RS crystal growth in the PLA channels is probably more randomly oriented than in thin ones, resulting in a depreciated effective *d*_33_.

Then, impedance spectroscopy was used to characterize each manufactured RS/PLA composite sample. The conductance and susceptance spectra (Figure 7) allowed us to identify the resonant behaviors of the piezoelectric samples and define their resonance frequencies (*f_r_*), quality factors (*Q*) and thickness coupling coefficients (*k_t_*). The average values and standard deviations of these parameters (Table 1) are shown for the two geometries of manufactured disks. Thicker (2.5 mm) samples have higher resonance frequencies ($f_{r}\;\approx$ 49 kHz) and quality factors (Q ≈ 80) than thinner (1.5 mm) samples ($f_{r}\;\approx$ 38 kHz and Q ≈ 37). This is probably due to the increased stiffness of thicker RS/PLA samples compared to thinner ones. In terms of coupling factors, thinner disks have better coefficients ($k_{p}\;\approx \;0.25$ for the 1.5 mm thick sample and 0.15 for the 2.5 mm thick sample).

## 3. Conclusions

In this work, we demonstrate the ferroelectric and piezoelectric performance of eco-friendly PLA/RS composites obtained by using additive manufacturing techniques. The samples were made by growing Rochelle salt crystals in a 3D-printed biodegradable PLA matrix with millimetric-sized meshes. An RS seed was attached to one side of the disk-shaped PLA matrix and used to allow the crystals to be oriented towards the matrix capillaries. Then, the direct and converse piezoelectric properties of the RS/PLA composite samples were characterized. Additionally, the ferroelectric behavior was illustrated below the Curie point at 24.9 °C. In terms of piezoelectric performance, we achieved an effective *d*_33_ of 30 pC·N^−1^ and 120 pC·N^−1^ on 2.5 mm and 1.5 mm thick PLA/RS composite samples, respectively. The transducers had a resonance frequency in the kilohertz range with quality factors ranging from 20 to 100. The *d*_33_ of manufactured samples was higher than most of commercial PVDF-TrFE films. We demonstrated that RS/PLA samples can easily be made at low cost in any fablab and with an eco-friendly approach. They have strong advantages in terms of cost, environmental impacts and energy budget, compared to most lead-based and lead-free piezoelectric components. This is partially due to process simplicity and the low amount of thermal energy required. The performances of such 3D-printed fully biodegradable piezoelectric composites could be of great interest to disposable harvester, sensor or actuator applications. In the future, the circular manufacturing of such piezoelectric salt-based components could be further developed to produce some components which limit the material intensity and minimize the environmental impacts. For the moment, this technology perfectly suits actual pedagogical needs and applications, illustrating the advanced physical or technological concepts and applications with a “low systemic environmental footprint” process in accordance with student requests. It was recently proposed that academic institutions make the urgent switch to content and concrete experiments that could help to prepare themselves for climate change challenges. In order to achieve this, “green” experimental matters and materials could be used to highlight industrially (more polluting) equivalent technological concepts or elements. Methods, materials and experiments presented in this work are in accordance with this eco-friendly academic transition, especially in the field of sensor and transducer development.

## 4. Experimental Section

### 4.1. PLA/RS Sample Manufacturing

First, PLA matrices were printed using Ultimaker 3D printer (V1.1) and PLA filament, also provided by the same company. In parallel, some RS crystal seeds were separately grown in order to obtain large single crystals that were at least 2 cm wide. The crystal growth was performed in an over-saturated solution at room temperature (around 21 °C) by mixing deionized water and RS powder provided by APC Pure Company (UK). Then, an RS seed was attached to a disk-shaped PLA matrix on one side. Notice that the “optical axis” (the longest tubular direction of crystals, Figure 1) of the seed must be parallel to the matrix plane (and electrodes). Having the optical axis perpendicular to the electrodes (parallel to electric field) would result in a weaker piezoelectric response. Then, the seed attached to the matrix deep in the solution is kept at 10 °C in order to initiate the liquid-phase crystal epitaxy through the PLA disk as shown in Figure 1. Once the PLA matrix was completely embedded in the growing RS crystal (after 36 to 48 h), it was removed from the solution. Then, the samples were naturally dried, and the excess RS was removed from the PLA matrix by cutting the RS crystal close to both sides of disk, and polished with sanding paper until the PLA matrix surfaces appeared again. Finally, electrodes were placed on the sample: two aluminum disks (10 µm thin) were cut into a commercial aluminum foil and affixed to both faces of the disk with Bare conductive ink (commercial name: “electric paint”). As mentioned previously, this manufacturing process is simple and feasible in any fablab without any sophisticated equipment. Both the energy and financial budgets required of such a process are low compared to standard piezoelectric ceramic or even organic samples, in terms of costs and complexity.

### 4.2. Experimental Setup and Characterization Tools

For each 3D-printed, fully biodegradable piezoelectric composite sample, we measured the impedance spectrum of the PLA/RS samples by using E4980A, Keysight impedance analyzer. The displacement spectra were acquired by employing an MSA-500 laser vibrometer from Polytech. The direct piezoelectric effect was evaluated with the setup described in Figure 8. This consisted of a pressure-controlled compressed air piston applying a controlled pressure to the sample, behind which was a sensor indicating the applied force to the sample. The output piezoelectric signal was measured on a scope (TDS2014C, Tektronix, Beaverton, OR, USA) loaded on 1 MΩ or 10 MΩ. The calculation of electrical charges (Coulombs) was performed using digital integration after noise-reduction of the signal.

## Figures and Tables

**Figure 1 materials-14-06132-f001:**
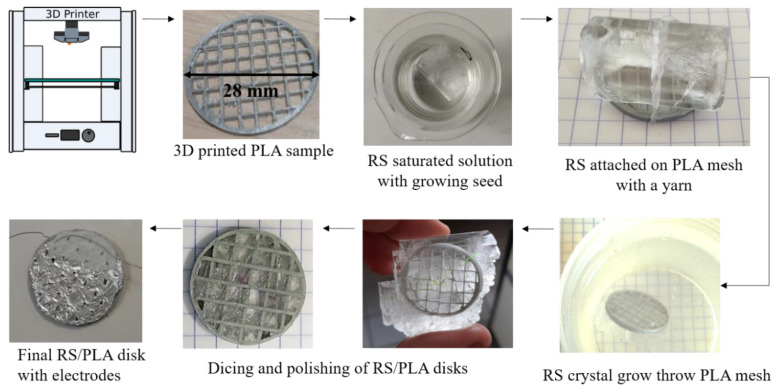
Manufacturing process flow of the piezoelectric samples.

**Figure 2 materials-14-06132-f002:**
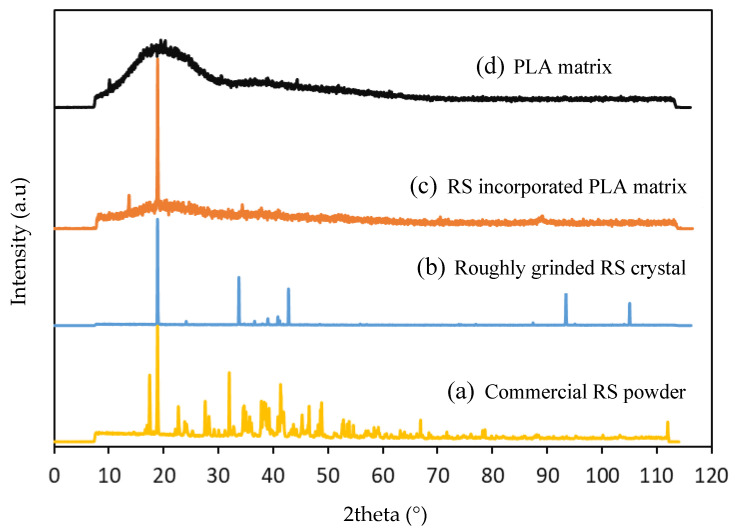
Comparative XRD diffractogram performed on (**a**) PLA matrix, (**b**) RS-incorporated PLA matrix, (**c**) roughly grinded single RS crystal, (**d**) commercial RS powder (provided by APC pure).

**Figure 3 materials-14-06132-f003:**
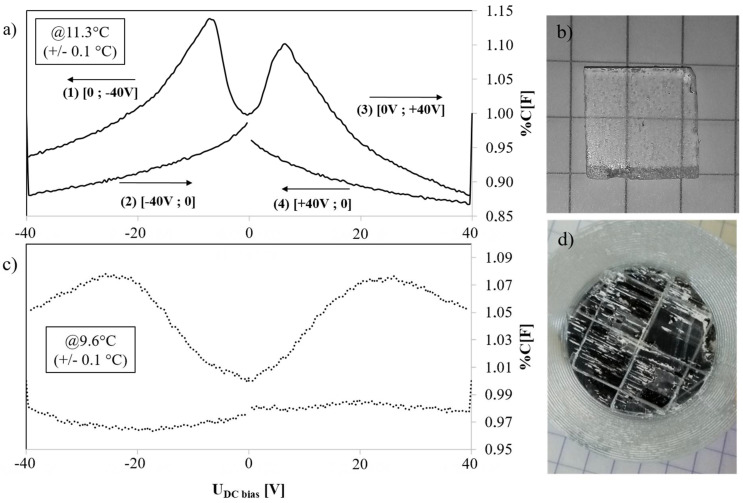
Normalized capacitance hysteresis loop measured on (**a**,**b**) thin RS square single crystal and (**c**,**d**) on RS/PLA thin sample, with bias voltage applied from −40 V to +40 V.

**Figure 4 materials-14-06132-f004:**
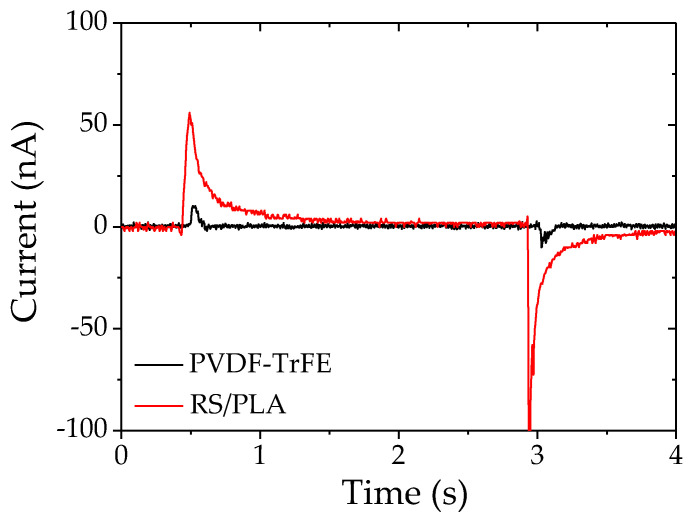
Harvested currents for one RS/PLA sample and the commercial PVDF-TrFE sample measured throw 1 MΩ input impedance under a pressure step of 0 to 1 bar.

**Figure 5 materials-14-06132-f005:**
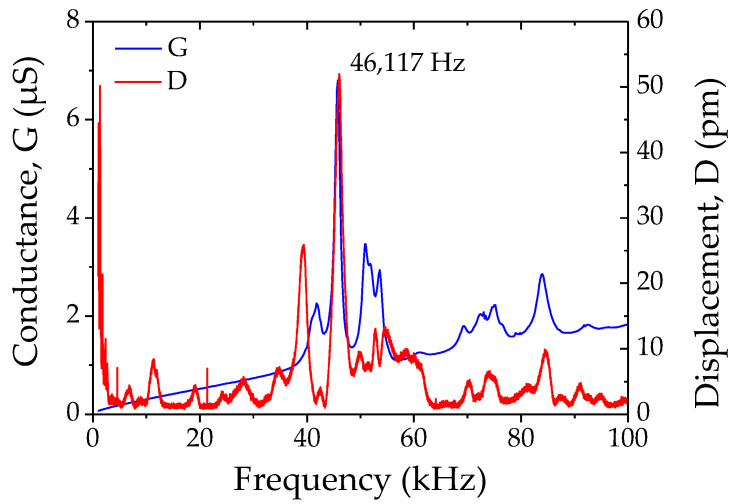
Conductance and displacement spectra vs. frequency recorded on the same sample (2.5 mm thick), respectively, using impedance spectrum (U = 1 V) and laser vibrometer (U = 2 V).

**Figure 6 materials-14-06132-f006:**
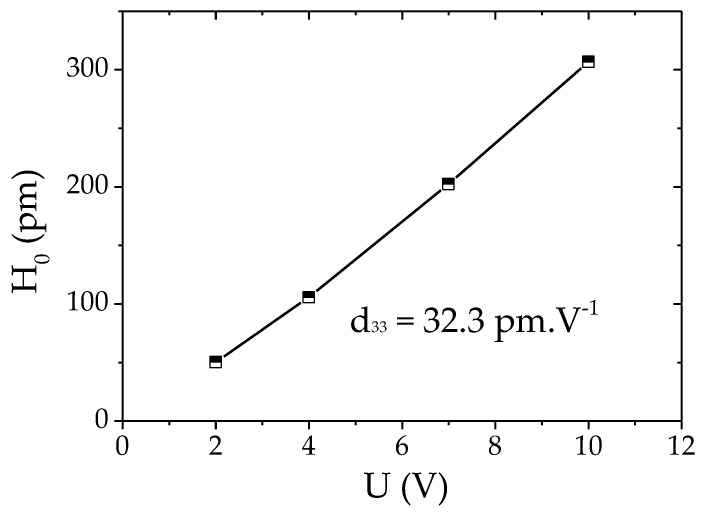
Static displacement (computed from vibrometer measurements) vs. applied voltage measured on one RS/PLA 2.5 thick sample.

**Figure 7 materials-14-06132-f007:**
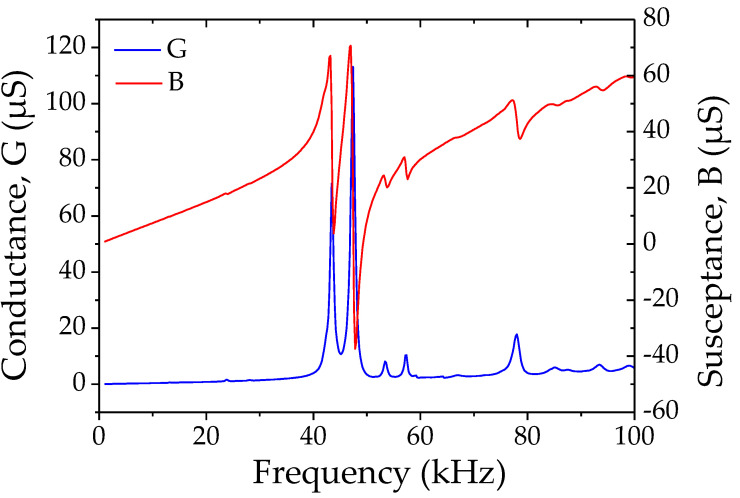
Conductance and susceptance spectra of 2.5 mm thick RS/PLA manufactured sample.

**Figure 8 materials-14-06132-f008:**
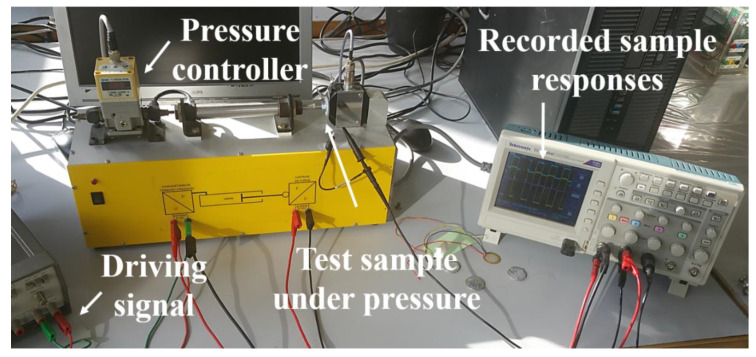
Dedicated direct piezoelectric coefficients measurement setup.

**Table 1 materials-14-06132-t001:** Average values (*μ*), and their corresponding standard deviations (*σ*) of direct piezoelectric coefficients calculated from charges harvested under a pressure step of 0 to 1 bar, converse piezoelectric coefficients calculated from static displacement and applied voltage, resonant frequencies (*f_r_* (Hz)), quality factors (*Q*), and thickness coupling factors (*k_t_*) extracted from impedance spectra. for two thicknesses (1.5 mm and 2.5 mm) of RS/PLA composite samples and for a commercial PVDF-TrFE sample. Thick ceramic (Pb(Mg_1/3_Nb_2/3_) O_3_-PbTiO_3_-PbZrO_3_) published data from the literature is also presented as reference [31]. NB: Rochelle salt (unstable) stated relative permittivity (*ε*_33_*)* = 9.5.

		Direct *d*_33_ (pC·N^−1^)	Converse *d*_33_ (pm·V^−1^)	Resonant Frequencies *f_r_* (Hz)	Q	** *k_t_* **
1.5 mm	μ	119	80	37,433	34	0.25
	σ	56	16	6226	13	0.04
2.5 mm	μ	55	32	49,132	83	0.16
	σ	19	14	3106	25	0.02
PVDF-TrFE	μ	22	NA	NA	NA	NA
	σ	1				
PZT [31]	*µ*	420	NA	NA	NA	0.52
		100				0.1

## Data Availability

The data presented in this study are available on reasonable request from the corresponding author.

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
