# Peer review of "Rochelle Salt-Based Ferroelectric and Piezoelectric Composite Produced with Simple Additive Manufacturing Techniques"

_materials, 2021, doi:10.3390/ma14206132_

Round 1
Reviewer 1 Report
Comments to the author
In the manuscript, the authors discuss ferroelectric and piezoelectric composites based on Rochelle salt. The article is interesting and well written. I recommend the paper for publication after a minor revision.
Minor suggestions (not mandatory, authors can decide):
1) Introduction - the authors wrote: "However, nowadays, piezoelectric organic ferroelectrics are attracted increasing attention due to their lower manufacturing environmental impact." I am a little confused. As I know, the organic ferroelectrics (at least some) can be very harmful to human health.
2) Unify the addressing of converse (sometimes in the text used also expression reverse) piezoelectric effect in the text.
3) Introduction, please correct µCm-2 and not mCm-2.
4) It would be interesting to examine images of ferroelectric domain structure performed with piezo-response force microscopy: http://dx.doi.org/10.1098/rspa.2018.0782. This would be another confirmation of ferroelectric behavior.
5) Please compare kp and Q values with those of good ferroelectric material in thick film form. For example, kp and Q values of Pb(Mg1/3Nb2/3)O3-PbTiO3 thick films.
Author Response
Article revision letter.
Ref: materials-1327575
Title: Rochelle salt based ferroelectric and piezoelectric composite produced with simple additive manufacturing techniques
Journal: Materials – MDPI
Thank you for reviewing this work. Please find here our response and the corrected manuscript attached (Please see the attachment):
1) As suggested by the reviewer, the sentence has been corrected (p.1): Nowadays, piezoelectric organic ferroelectrics are attracted increasingly attention, but some organic ferroelectrics can be harmful to human health.
2) reverse has been replaced by converse in the entire manuscript.
3) mCm-2 has been replaced by µC.m- 2.
4) The authors agree with the reviewer comment on the fact that PFM characterization could also be used to confirm the piezoelectric behavior of our material. In fact, we have already tried some PFM characterization on Rochelle salt, but did not obtain relevant results to show right now.
5) As proposed by the reviewer, in Table 1. we had thick PZT ceramic data from literature (p.6): Thick ceramic ( Pb(Mg1/3Nb2/3) O3-PbTiO3-PbZrO3 ) published data from literature is also presented as reference [31].

Reviewer 2 Report
This paper is a very instructive work worthy for publication after some minor revision:
In the introduction, I recommend to acknowlege also the work of A. Nicolson in 1918-1919 [A. McLean Nicolson, The piezoelectric effect in the composite rochelle salt crystal, A.I.E.E. Trans., v. 38, Part 2, pp. 1467-1485, 1919. DOI: 10.1109/T-AIEE.1919.4765643, Alexander M. Nicolson, Generating and transmitting electric currents. U.S. Patent No. 2,212,845A, filed April 10, 1918, issued August 27, 1940]. A more precise note on the instability of Rochelle salt in humid environment would be instructive for the reader.
Other shortages:
The paper containes a number of needless spaces.
In the measuring units pC·N-1 and pm·V-1 the multiplier sign should be corrected.
The journals should be formatted in Ref. 11,15, 16, 19, 25, 27, corrected in Ref. 21, 27, and unified for Smart Materials and Structures, cf. Ref. 4. Formate NbNaO3 in Ref. 11 and Rochelle salt in Ref. 20.
p. 3 mesured=> measured, literrature => literature
p. 5 Eigen frequency => eigenfrequency
The research in this paper is well and comprehensibly described. It does not provide new data confirming data measured about one century ago, but it considers the data under the aspects of green technology. Problems in measurement are well addressed.
figure 1 may be rearranged for a smaller width
Introduction should be extended by acknowledging the work of A. Nicolson. Also, a hint should be given to the low stability of Rochelle salt in humid environment. The latter is important for practical application.
Author Response
Article revision letter.
Ref: materials-1327575
Title: Rochelle salt based ferroelectric and piezoelectric composite produced with simple additive manufacturing techniques
Journal: Materials – MDPI
Thank you for reviewing this work. Please find here our response:
- needless spaces have been removed
- the measuring units pC·N-1 and pm·V-1 have been corrected.
- Journals references have been formated
- Measured and literature have been corrected (p.3)
- eigenfrequency has been corrected (p.5)
- Fig.1 has now smaller width
- In order to acknowledge the work of A. Nicolson, we add the following sentence (p.1) : One century ago, Nicolson designed Rochelle salt piezoelectric transducers[1].
- I have written a note in French where I mentioned Rochelle salt stability issues (https://hal.archives-ouvertes.fr/hal-03130122/document ). The aim of encasing RS into a matrix (and with top and bottom electrodes) is to strongly decrease these effects. This point has been mentioned in previous papers. I have “old” RS-based samples that still works after 4 years. It is not the case for unprotected RS crystals...
Reviewer 3 Report
In this paper, piezoelectric PLA-Rochelle salt composites have been fabricated and characterised. The advantages of this type of material lies in its recyclability and low temperature fabrication. Whilst this paper demonstrates an interesting fabrication technique that would be of interest to the wider piezocomposite community and seems to indicate some promising properties, there is some key information missing that is required to enable the work to be reproduced and I have some reservations about the accuracy of the experimental techniques used for the piezoelectric characterisation.
The paper needs a detailed method section that enables other researchers to reproduce this work; the flow diagram in Figure 1 is not really sufficient. The details of the PLA 3D printing process including equipment need to be given as do the conditions for the RS crystal growth including solution details, temperatures etc. Suppliers of seed crystals, electrodes, chemicals etc. should be given. Details of the equipment used to characterise the samples need to be listed.
The piezoelectric properties have been measured using a custom made set up whereby the piezoelectric current due to an applied force is measured. On page 5 it is stated that there is a large uncertainty in the data, which is given in table 1, but there is no indication of the number of samples fabricated and tested. The standard deviation seems very high and I wonder if this is related to the technique used to measure the piezoelectric coefficients. The 1 MOhm load resistance seems quite low, but then the results for the PVDF-TrFE seem reasonable. I wonder whether this equipment has been calibrated with any other materials, e.g. a commercial PZT, or alternatively, testing one of the composites on some commercial equipment (e.g. Berlincourt meter)? Details of the controlled load step would also be useful. Is there a reason why the PVDF-TrFE not been measured for its converse piezoelectric coefficent? This would provide a useful comparison and a feel for how accurate each measurement is.
Another issue related to the piezoelectric coefficients is that in the abstract the authors mention that this material hasn't been poled, despite the ferroelectric nature of RS being demonstrated with capacitance hysteresis measurements. Where do the piezoelectric properties arise from if the material isn't poled? Is the seed crystal orientated to give a preferential piezoelectric orientation of the RS?
It would be useful to report the permittivity of the RS and the composites as this is important for calculating the voltage sensitivity (g coefficient) and energy harvesting coefficient (d.g).
Impedance spectroscopy has been used to assess the resonant behaviour of the composites. The planar coupling coefficent, kp, has been reported it is unclear whether any modal analysis has been done to check that the resonance mode is the planar one (rather than the thickness mode, for example). The thicker sample has a higher resonance frequency, indicating it is significantly stiffer than the thinner sample, particularly as resonant frequency is inversely proportional to thickness. Is it possible to measure this in another way to check? It seems like there are too many unknowns to be confident that the correct mode has been calculated, particularly as the authors have stated that the thinner samples are not flat, which could induce other piezoelectric modes. It would help to detail the method by which kp and Q have been calculated.
A minor point but the first sentence of the abstract implies that piezoelectricity was first discovered using Rochelle salt - I think this should be piezoelectricity discovered in Rochelle salt (i.e. it had already been discovered but this was observed in a new ferroelectric material).
Author Response
Article revision letter.
Ref: materials-1327575
Title: Rochelle salt based ferroelectric and piezoelectric composite produced with simple additive manufacturing techniques
Journal: Materials – MDPI
Thank you for reviewing this work. Please find here our response:
In section 4 the manufacturing process including 3D printer brand, RS powder supplier, seed method, electrode commercial ink and measurement equipment are described.
Even if we didn’t have access to standard Berlincourt meter, our dedicated setup shouldn't give such dispersion of results (at least not on PVDF-TrFE or PZT sample tested in previous works). As mentioned into the paper, the dispersion of results is due to the random nature of crystallization. On some sample, seed orientation was conserved along all sample, but in the majority of cases a part (more or less important) of the sample had parasitic (random) crystallization that consequently decreased the performances of the transducers.
Piezoelectricity and ferroelectricity of Rochelle salt are natural properties of the crystal without any pooling required. Since RS has 3 non-neglectable orthogonal piezoelectric coefficients (d14, d25, d36), a piezoelectric response could be measured in almost every axes except the optical one. Thus, compared to e.g. ADP or KDP crystallography of RS is more complicated. So yes, generally speaking seed crystal could be orientated to give a preferential optimized piezoelectric response depending also on preferential mechanical solicitation. Including such level of detail with consistent demonstration of these physical properties would overpass the whole length of the paper. In addition, it has already been reported in the past. Useful information and experiments are provided for instance by Nicolson in 1919 on how to operate Rochelle salt transducers (Nicolson, A. M. (1919). The piezo electric effect in the composite rochelle salt crystal. Transactions of the American institute of electrical engineers, 38(2), 1467-1493. ) now cited in the manuscript.
In order to clarify our process and to respond to your request we have added the following precision (p.4):
Notice that the “optical axe” (the longest tubular direction of crystals, Fig. 1) of the seed must be parallel to the matrix plane (and electrodes). Having the optical axe perpendicular to the electrodes (parallel to electric field) would result in a really weak piezoelectric response.
Concerning mode shape, we have just performed Finite Element Method (FEM) simulations that suggest that it should be thickness modes as you suggested. So, thank you! We corrected the coefficient denomination (kt instead of kp) even if some incertitude exists on the effective orientation solicitated on our sample. Apart from calculation and simulation, for the moment we have no experimental tool to investigate more this point.
Reviewer 4 Report
- In regards of the lead-based and lead-free piezoelectric materials, recent important discussions on this topic should be added in the introduction part, such as:
Are lead-free piezoelectrics more environmentally friendly? MRS Communications, 2017, 7 (1), 1-7
Integrated hybrid life cycle assessment and supply chain environmental profile evaluations of lead-based (lead zirconate titanate) versus lead-free (potassium sodium niobate) piezoelectric ceramics, Energy & Environmental Science 2016, 9 (11), 3495-3520
BiFeO3-BaTiO3: A new generation of lead-free electroceramics, Journal of advanced dielectrics, 2018, 8 (06), 1830004
- recent works on 3D printed lead-free dielectrics should be shortly reviewed in the before starting your own work, such as:
Direct ink writing of bismuth molybdate microwave dielectric ceramics, Ceramic International, 2021, 47(6): 7625-7631.
Additively manufactured ultra-low sintering temperature, low loss Ag2Mo2O7 ceramic substrates, Journal of the European Ceramic Society, 2021, 41(1), 394-401.
Microstructure and microwave dielectric properties of 3D printed low loss Bi2Mo2O9 ceramics for LTCC applications, Applied Materials Today, 2020, 21: 100862.
Multi-Material Additive Manufacturing of Low Sintering Temperature Bi2Mo2O9 Ceramics With Ag Floating Electrodes by Selective Laser Burn-Out, Virtual and Physical Prototyping journal, 2020, 15(2): 133-147.
- XRD and SEM of the measured samples should be provided for the phase and micro structure discussion.
Author Response
Article revision letter.
Ref: materials-1327575
Title: Rochelle salt based ferroelectric and piezoelectric composite produced with simple additive manufacturing techniques
Journal: Materials – MDPI
Thank you for reviewing this work. Please find here our response:
We thank the reviewer for the many reference proposed. After reading them carefully, we decided to add the most relevant one (highlighted in yellow). The mention of previous piezoelectric 3D printed material has been added: Some of them could be 3D printed [18]
XRD of RS single crystal, RS powder, PLA and the measured samples have been now provided. The SEM imaging and analysis have been also performed and it is presented into supplementary information document since it doesn’t present highly important and relevant information.
Refs (highlighted in yellow => included):
Are lead-free piezoelectrics more environmentally friendly? MRS Communications, 2017, 7 (1), 1-7
Integrated hybrid life cycle assessment and supply chain environmental profile evaluations of lead-based (lead zirconate titanate) versus lead-free (potassium sodium niobate) piezoelectric ceramics, Energy & Environmental Science 2016, 9 (11), 3495-3520
BiFeO3-BaTiO3: A new generation of lead-free electroceramics, Journal of advanced dielectrics, 2018, 8 (06), 1830004
recent works on 3D printed lead-free dielectrics should be shortly reviewed in the before starting your own work, such as:
Direct ink writing of bismuth molybdate microwave dielectric ceramics, Ceramic International, 2021, 47(6): 7625-7631.
Additively manufactured ultra-low sintering temperature, low loss Ag2Mo2O7 ceramic substrates, Journal of the European Ceramic Society, 2021, 41(1), 394-401.
Microstructure and microwave dielectric properties of 3D printed low loss Bi2Mo2O9 ceramics for LTCC applications, Applied Materials Today, 2020, 21: 100862.
Multi-Material Additive Manufacturing of Low Sintering Temperature Bi2Mo2O9 Ceramics With Ag Floating Electrodes by Selective Laser Burn-Out, Virtual and Physical Prototyping journal, 2020, 15(2): 133-147.
XRD and SEM of the measured samples should be provided for the phase and micro structure discussion.

Round 2
Reviewer 3 Report
Thank you for your response. I have some minor comments that I believe would strengthen the paper if addressed, listed below.
Firstly though, apologies that I missed the experimental methods in section 4 on first review, I thought the reference to the Experimental section was solely to figure 1. Perhaps to avoid confusion you could add a reference to section 4 in the text in the first sentence of Results and Discussion on page 2, e.g. ‘…described in the Experimental section (see Section 4 and Fig. 1).’
Thank you for your explanation of the origin of piezoelectric properties in Rochelle salt. Presumably there is a dominant crystallographic orientation despite the randomly oriented crystals reducing the overall d33 value by virtue of it being grown rather than processed at high temperature like ferroelectric ceramics.
The d33 of a PZT ceramic has been added to table 1. This would provide a useful verification of the accuracy of the measurement system used (providing the measured d33 was close to the expected value – e.g. this could be done with a commercially produced piezoceramic with known d33 as a calibration). However, it’s not clear if this has been measured on this equipment or is just a value quoted from the literature; if it’s the latter, I do not really see the benefit of including in table 1. It would be useful to clarify this, and also clarify what the manufacture stated d33 value of the commercial PVDF is compared to the value you have measured.
As I mentioned in my previous review, the effective permittivity (ε33T) of a composite is useful for comparing the figures of merit of piezoelectric materials for both sensing (g33 = d33/ε33T) and energy harvesting (d33g33). The permittivity can be calculated from impedance spectroscopy at reported for example at 1 kHz (no need for new figure) and would be useful for other researchers. Similarly, I think it would help to give the equations from which quality factor and coupling factor were calculated. These comments have not been addressed.
Minor corrections -
‘Pooled’ and ‘pooling’ should be poled/poling as in change the polarity. ‘Axe’ should be axis
First sentence of abstract implies ferroelectricity discovered before piezoelectricity but should be other way round (piezoelectric effect discovered ca. 1880 by Curie brothers, before ferroelectricity, ca. 1920 by Vaselek)
Author Response
Article 2nd revision letter.
Ref: materials-1327575
Title: Rochelle salt based ferroelectric and piezoelectric composite produced with simple additive manufacturing techniques
Journal: Materials – MDPI
Dear Reviewer,
Thank you for revising again this work. Here are my point-by-point response :
- As suggested by the reviewer, reference to section 4 in the text in the first sentence of Results and Discussion on page 2, has been added on page 2.
the process described in the Experimental section (see Section 4 and Fig. 1).
- As suggested by reviewers, PZT reference value has been added for comparison. Values have been extracted from literature.
The commercial PVDF sample used as a stated d33 of 23pm/V. The following sentence has been added (p.6): . Notice that datasheet stated d33 is 23 pC·N-1.
- Concerning ε33, it is not so simple with Rochelle salt … As mentioned in [Slivka, A. G., et al. "Influence of external factors on dielectric permittivity of Rochelle salt: humidity, annealing, stresses, electric field." arXiv preprint cond-mat/0505439 (2005).], dielectric permittivity of Rochelle salt is not stable with respect to temperature (ambient range) and electric field (not strong field). But for model purpose here is the commonly stated values εr matrix (source efunda.com, cf. image.pdf): [image.pdf]
-
In order to give an order of value for relative permittivity as suggested, we add the following mention on table 1 (p.6) : NB: Rochelle salt (unstable) stated relative permittivity (ε33) = 9.5.
- ‘Pooled’ and ‘pooling’ have been replaced by poled/poling and ‘axe’ by axis
- Historical misunderstanding has been corrected (into abstract): More than one century ago, piezoelectricity and then ferroelectricity were discovered using Rochelle salt crystals

Reviewer 4 Report
accept
Author Response
We don't see any comment to respond here.